# The psychological impact of major disasters on Japan's medical system: An SNS text analysis

**Tomoya Kitayama** ◉ *, **Kanae Nishimura**

School of Pharmacy and Pharmaceutical Sciences, Mukogawa Women's University, Nishinomiya, Hyogo, Japan

* tomokita@mukogawa-u.ac.jp

## Abstract

A major disaster creates an extraordinary situation for medical system. The aim of this study is to evaluate the impact of infrastructure damage caused by a major earthquake on the medical systems in Japan. In this study, we analyzed the content of X (formerly Twitter) to assess the impact of the Noto Peninsula Earthquake that occurred on January 1, 2024 on the medical system, including psychological aspects. Posts including the term "prescription records" were compiled from October 2023 to March 2024. ML-Ask, a python library, was used to evaluate the emotional register of the post basis on ten elements: Joy, Fondness, Relief, Gloom, Dislike, Anger, Fear, Shame, Excitement, and Surprise. The presence of earthquake-related terms was prominent in tweets about prescription records in January, but not in February or March, 2024. The analysis found that in January, words associated with negative emotions were related to the disaster, infrastructure and medical system. From the results of the hierarchical cluster analysis of posts in January, the primary factor triggered negative emotions during the earthquake was the unreliability of electronic medical systems following the loss of power supply. These results suggest that during the earthquake in Japan, a significant number of people harbored negative sentiments toward electronic medical systems and questioned their reliability in disaster situations.

## Introduction

One of the medical systems in Japan is a system in which patients manage their prescription records in booklet form. In Japanese, this booklet is called "Okusuri Techo," which directory translates to "medicine record book." This system has been covered by insurance since 2000, and is actually operated in the booklet carried by the patient and used by the pharmacist at the dispensing pharmacy to record prescribed drug information. In addition to the chronological drug record, it is required to contain a record of the patient's name, date of birth, contact information, and other patient details; a record of the basis for drug therapy, such as allergies and adverse drug reactions; and a record of major medical history. Based on this information, the

**Data availability statement:** All relevant data for this study are publicly available from the GitHub repository (https://github.com/tomokitamukogawa/Psychological-impact-of-disasters-on-system).

**Funding:** The author(s) received no specific funding for this work.

**Competing interests:** No authors have competing interests.

pharmacist reviews the prescriptions with the main purpose of preventing side effects and avoiding drug duplication.

With regard to pharmaceutical information, digitization has been promoted in recent years. In particular, the Japanese government is promoting digitization using the Individual Number (as nicknamed "My Number") system. Patients can check drug information based on insurance claim information from Mynaportal, as online site for government services linked to My Number, via a dedicated app. In addition, dedicated apps for electronic prescription records are beginning to be used. Electronic prescriptions are considered necessary for sharing patient information not only among pharmacists but also among physicians and other healthcare professionals, and are being promoted in accordance with the Japanese government's policy of digitizing medical information. In the past, many researchers have studied changes in the prescription record system, prevalence, problems in use, and digitization [1,2]. However, the impact of a major disaster on prescription record systems is unknown.

In the event of a major disaster, hospitals may lose their functions due to power outages, building collapses, fires, tsunamis, etc. In terms of transportation, medical institutions may not be able to provide medical care due to damaged transportation systems and roads. In addition, medical records, including electronic files, are likely to be lost, making it impossible to access patient information. Doctors working in affected areas can cope to some extent with victims who have urgent needs such as trauma, but many problems arise when coping with those who have chronic illnesses. This is caused by the patient's inability to receive medical care from the family doctor or the doctor's inability to confirm the patient's staple medication due to the loss of medical records. In fact, the inability to diagnose and prescribe occurred after the Great Hanshin-Awaji earthquake. In response to this experience, an exception to the law has been in place since 2005, which allows pharmacists to prescribe drugs at their discretion if they have verified the prescription record in the booklet. Thus, in the event of a disaster, victims who have chronic illnesses can now receive necessary medications if they have prescription records. This exception was also applied in the Noto Peninsula Earthquake that occurred on January 1, 2024, with a maximum seismic intensity of 7 [3], equivalent to the revised Mercari Seismic Intensity Levels VIII, and a magnitude of 7.6 (16 km below the epicenter) [4], and three-component composite maximum acceleration 2,828 gal (28 m/s$^2$) [5]. Since the tip of the peninsula was the center of the affected area, lifelines were severely damaged, including massive power outages, water outages, and road disruptions.

This study examined posts on X (formerly Twitter) in the three months before and after the Noto Peninsula Earthquake that occurred on January 1, 2024, and analyzed the psychological changes of the Japanese public toward the medical care system, focusing on prescription records.

## Methods

### Analysis target

The analysis target included posts on X for a six-month period from October 1, 2023 to March 31, 2024. The Noto Peninsula earthquake occurred on January 1, 2024. X

was selected as the subject of analysis due to its widespread usage across diverse generations and its high penetration rate in Japan [6]. While other platforms such as LINE, YouTube, Instagram, and TikTok also maintain large user bases in Japan, they were excluded from this study. LINE was omitted due to its nature as a private communication tool, which presents privacy constraints. The other platforms are primarily video- or image-oriented, rendering them unsuitable for the text mining methods employed in this research. The X post data were collected using Python 3.8.5 (https://www.python.org/). Python-based libraries used were tweepy, schedule, pytz, and pandas. Using Python,.csv data was generated (https://github.com/tomokitamukogawa/Psychological-impact-of-disasters-on-system) [7]. The program was implemented by the authors, who rotated roles every one weeks. It should be noted that due to limitations of the Twitter API (X API Basic) and Internet environment, not all relevant posts were collected. Data collection was automated using a Python script, ensuring that the process was mechanical and independent of the authors' subjective intent. Likewise, if there were failures in the collection process, they were mechanical problems and occurred randomly, independent of the authors' intentions. In this study database, there were 38,567 posts containing the term "medicine record book." We confirm that the collection and analysis method complied with the terms and conditions for the source of the data.

## Ethical considerations

Research using social networking service (SNS) data has increased in recent years, but uniform research ethics guidelines around it are yet to be developed [8]. According to Twitter's terms of service, when users register, they are required to allow third parties to use the content they post. In fact, since the first post was published when the company was founded in 2006, all public posts are searchable on the Internet. In addition, Twitter revised its developer policy in 2020, clearly stating that it is possible to collect posted content for non-commercial research purposes, and clarifying the rules for using academic data. Therefore, in the case of big data analysis, the data is public information, and it is often argued that ethical review is unnecessary.

In recent years, ethical guidelines for research using SNS data have been proposed [9,10]. Most notably, George Washington University has published research ethics guidelines for using SNS data (George Washington University Libraries, online). The contents of the guidelines are as follows: the rules of the SNS platform must be followed, collection should be limited to public data, consent should be obtained when using data such as direct messages, collection of metadata such as profile information should be kept to a minimum, consent is required when quoting and posting the text of SNS, and user IDs and account names may only be provided when consent has been obtained. In this study, data was managed according to these guidelines. Therefore, the SNS text, user ID, and other identifying data is not described in the text, nor is it provided to third parties.

## Analysis of collected posts

The collected posts were analyzed using KH Coder (https://github.com/ko-ichi-h/khcoder) [7,11–13], and Python 3.8.5. The Python-based library used was ML-Ask (https://github.com/ikegami-yukino/pymlask) [7,14]. KH Coder is a software for quantitative text analysis or text mining, and was used for correspondence analysis and hierarchical cluster analysis. KH Coder is written in Perl, and uses ChaSen, MeCab, TermExtract, MySQL, R language and others as backends. ChaSen, MeCab, and TermExtract are morphological analysis engines, MySQL is a relational database management system, and R is a statistical programming language.

The Python library, ML-Ask, was used to perform the sentiment analysis of the posts. ML-Ask performs morphological analysis of input text, and compares it to an emotion dictionary database to classify it into 10 axes: "Joy," "Fondness," "Relief," "Gloom," "Dislike," "Anger," "Fear," "Shame," "Excitement" and "Surprise." Contextual emotion was also estimated based on Contextual Valence Shifters. For example, in the case of the sentence "I can't say I like it," ML-Ask infers that it is "Dislike," the opposite emotion to "like," because "like" is negated [7,14]. The resulting data were generated as.csv files using Python. Some of the posts contained URL and tag (ex; @aaaaa) information that had no meaning in Japanese, and

preprocessing was performed to remove them, which prevented analytical errors [7]. Since the trial analysis by KH Coder detected words separated by inappropriate delimiters, forced extraction words were set (S1 Text).

Dataset-specific validation of ML-Ask was performed using random sampling. We determined the minimum sample size using the Raosoft sample size online calculator (http://www.raosoft.com/samplesize.html; Raosoft Inc., Seattle, WA, USA) based on a total population of 38,567 posts, assuming a 5% margin of error and a 95% confidence level. Although the required sample size was 381, we randomly selected 400 samples using a Python script (https://github.com/tomokitamuk-ogawa/Psychological-impact-of-disasters-on-system) to ensure robustness.

Each of the 400 extracted sentences was evaluated independently by the authors. Disagreements were initially addressed through discussion to determine the final ground truth label. However, in cases where a consensus could not be reached, both sentiments identified by the authors were adopted as valid ground truth labels. Since ML-Ask supports multi-label classification, we constructed a 2×2 contingency table for each emotion and calculated the Cohen's kappa coefficient for each, as well as the macro-average (S1 Table). All calculations were performed using Python (https://github.com/tomokitamukogawa/Psychological-impact-of-disasters-on-system).

The strength of agreement was interpreted according to the criteria of Landis and Koch (1977): Slight (0.00–0.20), Fair (0.21–0.40), Moderate (0.41–0.60), Substantial (0.61–0.80), and Almost Perfect (0.81–1.00) [15]. The macro-average kappa coefficient across all emotions was 0.449, indicating "moderate" agreement [16]. Notably, Anger, Dislike, Fear, and Shame showed particularly high agreement, with kappa coefficients exceeding 0.6 (S1 Table). It should be noted that emotions with lower prevalence tended to yield lower kappa coefficients.

The number of occurrences of each emotion word detected in posts for a given time period was counted. To account for variations in the total number of posts across different periods, we calculated the percentage of posts containing each specific emotion. The percentage was defined as: (number of posts with a specific emotion/ total number of posts in the period) × 100. The study period was divided into four intervals: October to December, January, February, and March. Differences in the ratios for each time period were tested for significance using residual analysis. The association coefficient (Cramer's V) was calculated as the effect size [17]. The residual analysis was performed using the following formulas:

$$Expected\ value\ (E_{ij}) = (\sum_{i=1}^{a} n_{ij} \times \sum_{j=1}^{b} n_{ij}) / \sum_{i=1}^{a} \sum_{j=1}^{b} n_{ij}$$

$$Residuals\ variance\ (R_{ij}) = (1 - \frac{\sum_{i=1}^{a} n_{ij}}{\sum_{i=1}^{a} \sum_{j=1}^{b} n_{ij}}) \times (1 - \frac{\sum_{j=1}^{b} n_{ij}}{\sum_{i=1}^{a} \sum_{j=1}^{b} n_{ij}})$$

$$x_{ij} = (n_{ij} - E_{ij}) / \sqrt{E_{ij} \times R_{ij}}$$

$$p\ value = 2 \times (1 - (\frac{1}{\sqrt{2\pi}} e^{-\frac{x_{ij}^2}{2}}))$$

Cramer's V was calculated by the following formulas:

$$Chi\ square\ value\ (X^2) = \sum_{i=1}^{a} \sum_{j=1}^{b} \frac{(n_{ij} - E_{ij})^2}{E_{ij}}$$

 

$$\text{Cramer's V} = \sqrt{X^2 / (\sum_{i=1}^{a} \sum_{j=1}^{b} E_{ij} \times (\text{number of categories } - 1))}$$

$n_{ij}$ indicates the value of each cell. (i;columns, a is the maximum value. j;rows, b is the maximum value.).

The number of categories was determined based on the smaller dimension. Therefore, a value of 4, corresponding to the number of periods (columns) in Table 1, was selected rather than the number of emotions (rows).

Multiple comparison tests for the frequency of posts containing each emotion were performed using GraphPad Prism 6.04 (GraphPad Software, San Diego, CA, USA). Data are presented as the mean ± standard deviation (SD). The frequency of posts was analyzed using a two-way analysis of variance (ANOVA) with Emotion and Period as independent factors. Following the ANOVA, a Tukey-Kramer post-hoc test was conducted to identify specific differences between groups. A p-value of < 0.05 was considered statistically significant.

We focused our evaluation of ML-Ask's sentiment recognition on three specific emotions: Fear, Dislike, and Anger. These categories were selected due to their high prevalence in detection and strong agreement with human judgment, as evidenced by high kappa coefficients. Correspondence analysis was performed using period elements as external variables, and words highly related each period were extracted. This correspondence analysis was performed using the KH Coder software to obtain a more objective classification that was not biased by the researcher's viewpoint. Correspondence analysis is a multivariate statistical technique that visualizes a cross-table and displays the relationship between the number of occurrences of a word and a group (period elements) as a distance. The distance is calculated by dividing the Euclidean distance by the square root of the ratio to the total number of words appearing in each group. For this purpose, the entire profile was placed at the origin (0,0) and the feature was judged by the direction from which

**Table 1. Percentage of posts in each emotion.**

| Emotion (Mean±S.D.) | October–December | January | Februry | March |
|---|---|---|---|---|
| Fear (12.20%±0.009269) <br> $P < 0.05$ v.s. All other emotions | 12.56% | 10.86% | 12.41% | 12.98% |
| Dislike (5.98%±0.003621) <br> $P < 0.05$ v.s. emotions other than anger | 5.73% | 5.60% | 6.27% | 6.30% |
| Anger (4.01%±0.003095) <br> $P < 0.05$ v.s. emotions other than dislike, shame, surprise | 3.81% | 4.45% | 3.98% | 3.78% |
| Shame (2.92%±0.003021) <br> $P < 0.05$ v.s. emotions other than anger, surprise, relief | 2.59% | 3.14% | 2.74% | 3.21% |
| Relief (2.23%±0.002117) <br> $P < 0.05$ v.s. emotions other than shame, surprise, excitement, joy | 2.39% | 2.19% | 1.95% | 2.40% |
| Surprise (1.86%±0.00388) <br> $P < 0.05$ v.s. emotions other than gloom, shame, anger, fondness, excitement, relief, joy | 2.04% | 1.33% | 1.82% | 2.23% |
| Joy (1.59%±0.002836) <br> $P < 0.05$ v.s. emotions other than gloom, surprise, relisf | 1.44% | 2.01% | 1.46% | 1.43% |
| Gloom (1.37%±0.001962) <br> $P < 0.05$ v.s. emotions other than surprise, excitement, joy | 1.37% | 1.61% | 1.13% | 1.35% |
| Excitement (1.22%±0.002662) <br> $P < 0.05$ v.s. emotions other than gloom, surprise, relief, joy | 0.98% | 1.60% | 1.13% | 1.17% |
| Fondness (0.46%±0.000635) <br> $P < 0.05$ v.s. emotions other than surprise | 0.39% | 0.54% | 0.43% | 0.46% |
| Total | 16,974 | 9,763 | 5,327 | 6,503 |

$P$ values compared other emotions values, as determined by Tukey-Kramer test.

Note: The emotion recognition engine detects multiple emotions in a single sentence. The table aggregates all detected emotions.

the word appeared. KH Coder was also used for hierarchical cluster analysis. Co-occurring words were classified from the content of the posts, highly related words formed clusters, and trends in the posted content were inferred from the results.

## Results and discussion

### Trends in emotions over the entire period

The period was divided into three parts: the three-month period before the Noto Peninsula earthquake was considered the normal period; January was the immediate post-earthquake period; and February and March were the post-earthquake system restoration period. Posts containing specific emotions during each period were tabulated (Table 1). Cramer's V was calculated to clarify how each emotion changed over time and was found to be very low, at 0.06383. Since Cramer's V values of 0.15 or more are considered relevant [17], it was strongly suggested from the present results that there was no relationship between the period before and after the earthquake period and emotion, and there was no significant change in the proportion of posts containing each emotion throughout the analyzed period. From these results, it would appear that a major disaster does not elicit specific emotions toward the medical system.

In comparisons between emotions, "Fear" was detected at a significantly higher rate than all other emotions. The next most frequent emotions were "Dislike" and "Anger." For posts containing "Dislike," a significant difference was found for all other emotions except "Anger," and for "Anger," a significant difference was found for all other emotions except "Dislike," "Shame," and "Surprise." "Shame," which was detected at the next highest rate, had a significantly lower occurrence than "Dislike," one of the top three emotions with the highest rate of detection, and is considered to be one step lower than the second group formed by "Dislike" and "Anger." As a result, negative emotions such as "Fear," "Dislike," and "Anger" were detected more frequently for posts that included prescription record systems. "Fear" dominated the other emotions, followed by "Dislike" and "Anger" as the next most common group. Since "Shame" was significantly less common than "Dislike," subsequent analyses were performed on "Fear," "Dislike," and "Anger."

### Trends in posts for each emotion over time

Correspondence analysis was used to analyze the relationship between the posts in which "Fear," "Dislike," and "Anger" were detected and the corresponding time periods. The analysis results for all these emotions showed that they formed groups during the normal period from October to December, the immediate post-earthquake period in January, and the post-earthquake system restoration period from February to March, respectively (Fig 1). This suggests that the earthquake clearly triggered negative emotions in different areas of concern compared to the normal period from October to December. Also, the trend in the content of posts was different between the normal period of October-December and the post-earthquake period of February-March.

In those posts in which "Fear" was detected after the earthquake, words such as blood pressure, health, management, diagnosis and pain were detected in the normal period from October to December, and health-related content was often found (Fig 1A). In the immediate post-earthquake period, words related to the earthquake such as "disaster" and "evacuation," and words related to the prescription management system, such as "paper," "smartphone," "app," and "My Number," were detected. Meanwhile, in February and March, they formed a group and were related to medical institutions such as "medicine," "doctor," "prescription" and "pharmacy". Similarly, in the posts containing "Dislike," from October to December and February to March, words related to medical institutions were mainly detected, but in January, there were words related to the earthquake such as "disaster," "power outage" and "evacuation", as well as those related to prescription management systems such as "My Number" and "analog" (Fig 1B). For "Anger," from October to December, medical and system-related words such as "diagnosis," "photograph," "all at once" and "medical record" were detected, whereas in January, there was a strong association with the earthquake-related words such as "disaster," "evacuation" and "escape". In February and March, "medicine" and "pollen" were detected (Fig 1C).

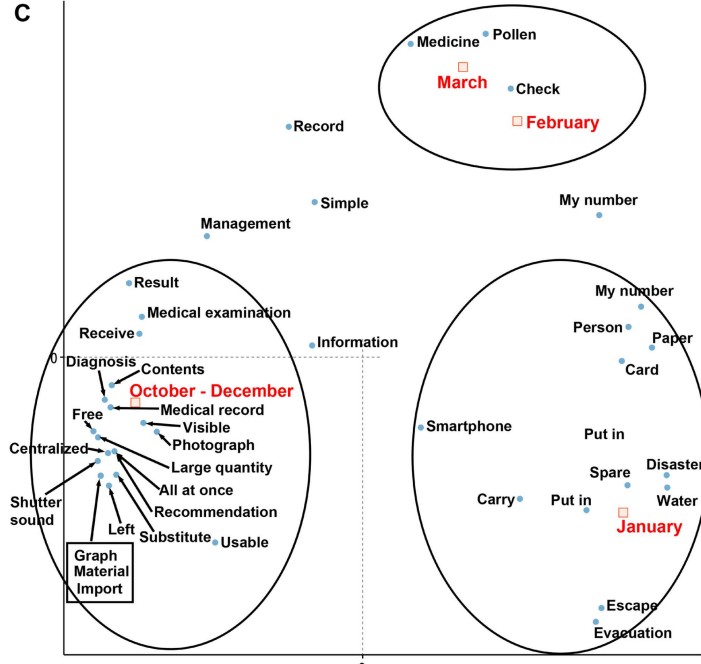

**Fig 1. Correspondence analysis of posts during the period from October 2023 to March 2024.** Correspondence analysis shows the relationship between each period. The black circles highlight areas of strong correlation with each period. Each figure indicates the results for the following: A: the analysis of posts related to fear, B: the analysis of posts related to dislike, C: the analysis of posts related to anger. The annotations for conversion from Japanese are shown. My Number includes Japanese abbreviations.

The analysis of each emotion suggests that the impact of the earthquake subsided in January, and that people's mental state had returned to quasi-normality by February. One possible reason for the difference in the analysis results between the normal period (October-December) and the post-earthquake period (February-March) is the seasonal change. For example, in the analysis of "Anger," "pollen" was detected in March. This is related to pollen allergies, which are most prevalent in Japan in February-March.

## Changes in posts regarding "Fear" over time

From the results of the hierarchical cluster analysis of posts in which "fear" was detected, we focused on clusters containing words detected by correspondence analysis (Fig 2, S1 Fig). In posts from October to December, "blood pressure," "systolic," "diastolic," "heart rate," and "app" formed a cluster. Booklets and apps that manage prescription records may have a function to record daily blood pressure. Thus, it can be inferred that the emotion of "Fear" was detected in relation to the function of being able to manage one's own health or condition (Fig 2A). Likewise, "medical examination," "management," "explain," "diagnosis" and "health" formed a cluster. The same cluster included words such as "information,"

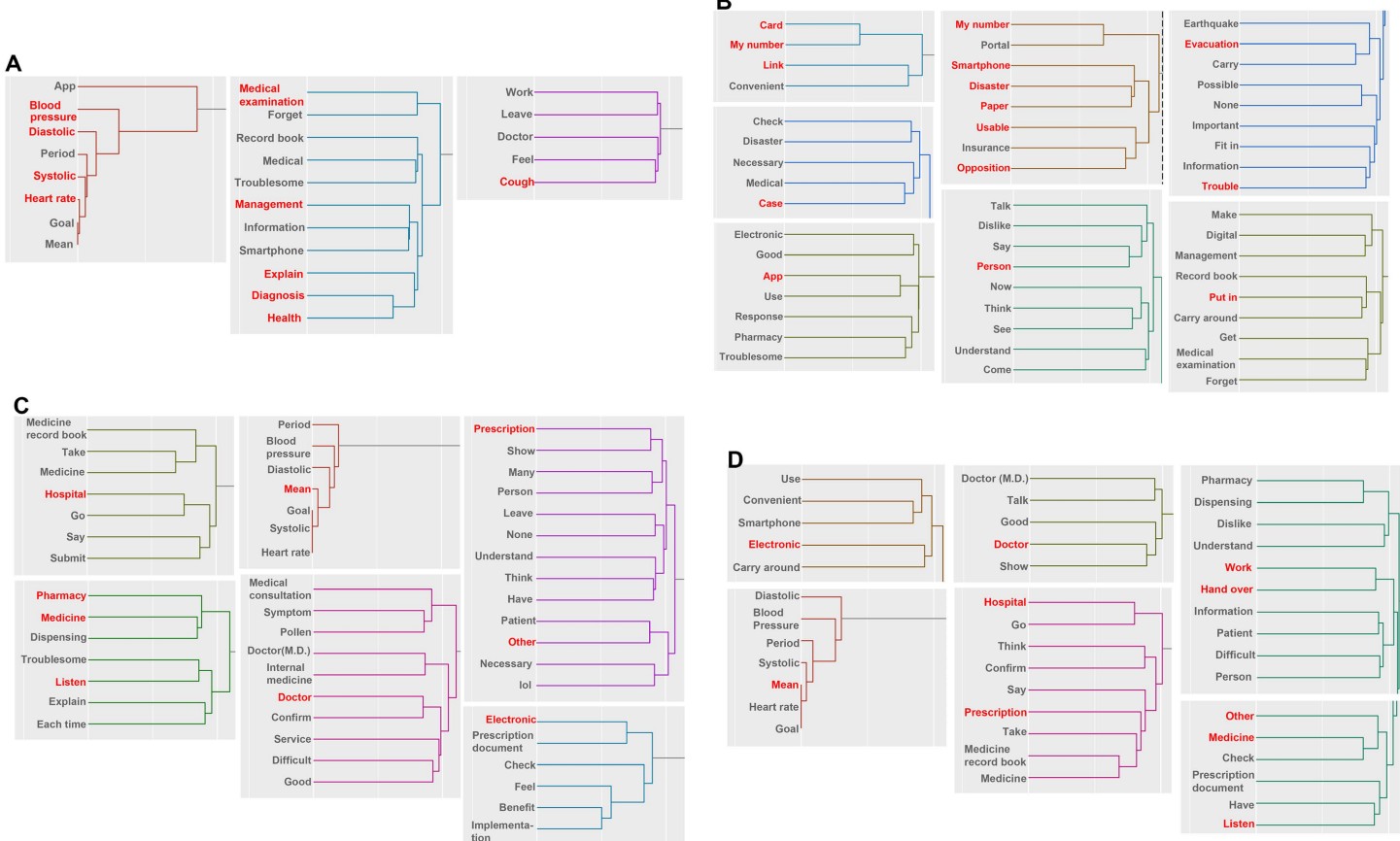

**Fig 2. Hierarchical cluster analysis of posts in which fear during the period from October 2023 to March 2024.** Each clusters shows a strongly related section in the correspondence analysis of posts related to fear. Each figure indicates the results for the following periods: A: October to December, B: January, C: February and D: March. The annotations for conversion from Japanese are shown below. The My Number includes Japanese abbreviations. The meaning of Medicine record book is A booklet used to keep a patient's prescription history, and lol (laughing out loud) is a translation of Japanese internet slang used to express laughter into English internet slang.

"smartphone," "record book," "troublesome" and "forget," suggesting that people feel "Fear" due to the hassle of managing health information on smartphones or booklets, or forgetting records or forms. As for "cough", it is thought that people were afraid of the impact of coughing due to their work or relationships with doctors.

Two clusters related to My Number were detected in the posts for January (Fig 2B). The cluster formed by "card," "My Number" and "link" also included the word "convenience." This suggests that "Fear" was detected in relation to linking the My Number Card to other medical systems. Since the card is equipped with an IC chip, and information can be read using facial recognition or a PIN, it is assumed that this stems from security concerns. As for "convenient," there was apparently a feeling that convenience was lost during the earthquake. This is supported by the content of another cluster, the cluster including "My Number," "smartphone," "disaster," "paper," "usable" and "opposition." It should be noted that KH coder aggregates Japanese verb and adjective conjugation forms. Thus, the present tense word "usable" in this context includes all past tense, future tense, and negative forms. For this reason, an additional analysis of the word "usable" was performed. This revealed that 68.5% of the "usable" extracted in the analysis was used in the posts as the negative form "unusable," which was highly relevant to the electronic medium of the My Number Card. The emotion of "Fear" was, therefore, evidently due to the inability to use electronic media such as Mynaportal or smartphones in the event of a disaster. Also, as regards words detected by correspondence analysis, there were respective clusters that included "case," "evacuation," "trouble," "app," "put in," and others. Therefore, it can be inferred that prescription records are important when evacuating in the event of a disaster, and it is necessary to take the records with you and carry them with you at all times. There were probably many posts related to disaster-preparedness throughout Japan, and it is possible that the emotion of "Fear" was linked to a sense of crisis regarding the disaster itself. In the analysis for February and March, a cluster of posts considered to be about prescription records, prescriptions, and especially the benefits and use of electronic prescriptions were associated with the results of the correspondence analysis (Fig 2C, D).

### Changes in posts regarding "Dislike" over time

In the cluster analysis for October to December, words that were highly associated with the correspondence analysis were mainly related to the use of prescription records (Fig 3A, S2 Fig). The words "listen" and "doctor" detected in the correspondence analysis formed one cluster, indicating that patients felt "Dislike" to showing their prescription records and listening to explanations from doctors. There also seemed to be an aversion to forgetting to write down prescription records during consultations, going to the hospital, and managing and writing down unnecessary information. However, from this analysis, the relationship between the words "thank you" and "Dislike" was unclear.

In the analysis of posts for January, "Dislike" was detected in relation to the insurance function of the My Number Card (Fig 3B). This can be inferred from another cluster, including "when," "recovery," "analog," "handle," "priority," "power outage," "clear" and "disaster." The nature of the words in this cluster suggests that the posts are about the fact that electronic prescription records cannot be used in a power outage due to a disaster, and that analog, i.e., paper-based booklet prescription records, are superior. It also appears to be strongly related to the frustration of not knowing when the communication infrastructure will be restored. This cluster includes "criticism,"and "Kono" was detected, which is the name of the Minister of Digital Affairs who promoted medical digitization at the time and recommended the My Number Card. This suggests that the content was critical of the minister in charge regarding the use of the My Number Card during disasters. In the analysis for February and March, as with the analysis of "Fear," a cluster of posts apparently related to the use of prescription records was associated with the results of the correspondence analysis (Fig 3C, D).

### Changes in posts regarding "Anger" over time

In the cluster analysis from October to December, words that were strongly associated with the correspondence analysis mainly concerned the specifications of electronic prescription records (Fig 4A, S3 Fig). This could be "Anger" at the lack of implementation of features related to the content of electronic media prescription record specifications. It could also be

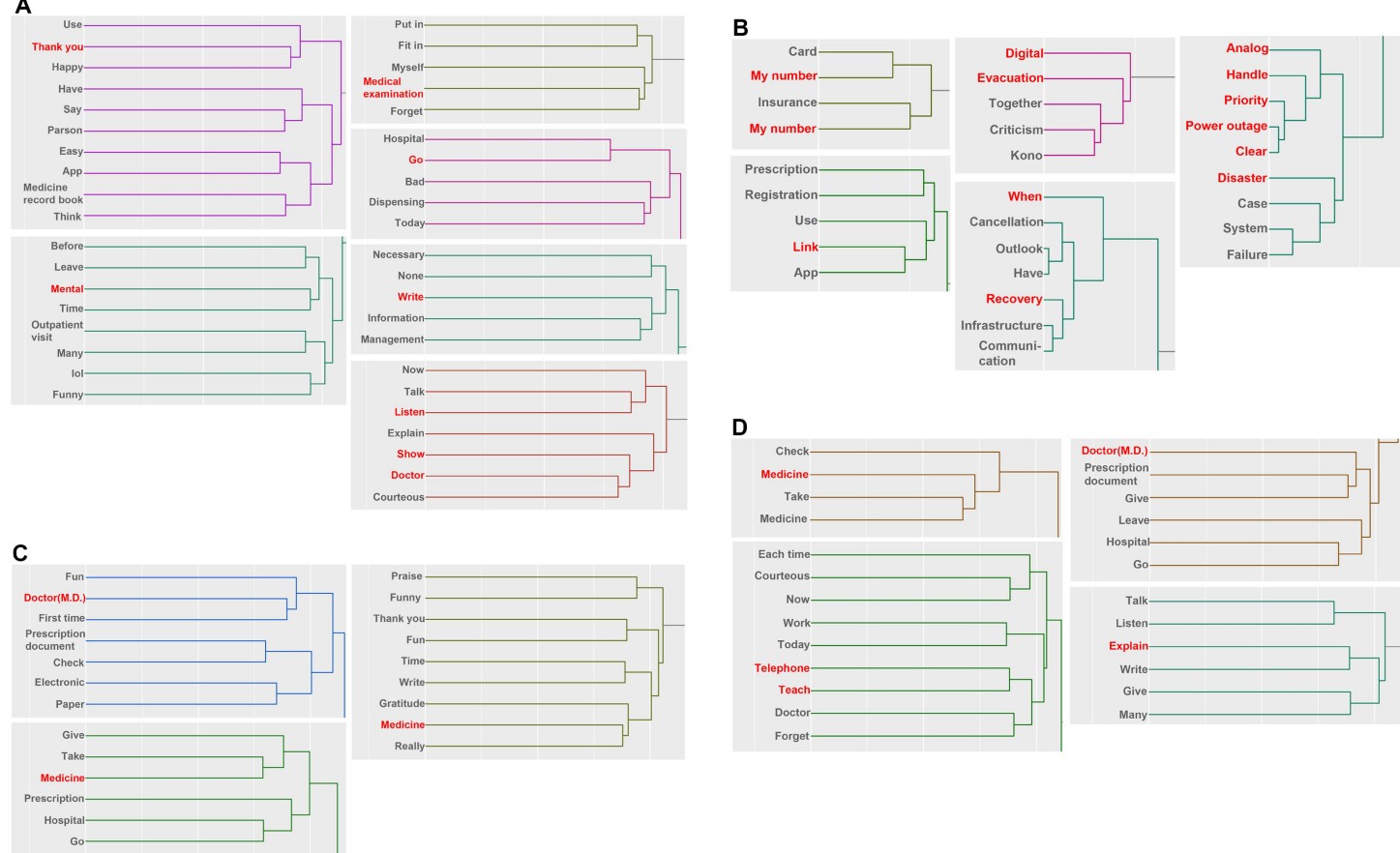

**Fig 3. Hierarchical cluster analysis of posts in which dislike during the period from October 2023 to March 2024.** Each clusters shows a strongly related section in the correspondence analysis of posts related to dislike. Each figure indicates the results for the following periods: A: October to December, B: January, C: February and D: March. The annotations for conversion from Japanese are shown below. The My Number includes Japanese abbreviations. The meaning of Medicine record book is A booklet used to keep a patient's prescription history, and lol (laughing out loud) is a translation of Japanese internet slang used to express laughter into English internet slang. The word of medicine in (D) is duplicated, but this is because they are listed using different Japanese words that have the same meaning.

due to comparisons with booklets, based on the words "visible," "large quantity," "all at once," "can be captured," "graphs" and "material". When recording prescriptions on a smartphone, it is difficult to see a large volume of data all at once due to screen display issues. Moreover, when it comes to capturing data, booklet prescriptions can simply be copied, but in the case of an electronic version, security settings to be made before the data can be transferred, thus medical facilities without the proper equipment choose to transfer handwritten records.

Analysis of the posts for January detected "Anger" regarding behavior during disasters, especially evacuation (Fig 4B). This is thought to be anger at not having water and staple medications, and not having the time to take medications with them when evacuating. In addition, the clusters analyzed suggest that in the event of a disaster or emergency, there is a need to carry medications and their information, and a need to keep prescription records in a wallet or similar. In the analysis for February and March, as with the analysis of "Fear," clusters of posts apparently related to the use of prescription records were associated with the results of the correspondence analysis (Fig 4C, D). "Pollen" was specifically detected in the correspondence analysis for "Anger," suggesting that the "Anger" was over hospital visits due to pollen allergies, the association with the psychological impact of the earthquake was low, and it was actually due to the season.

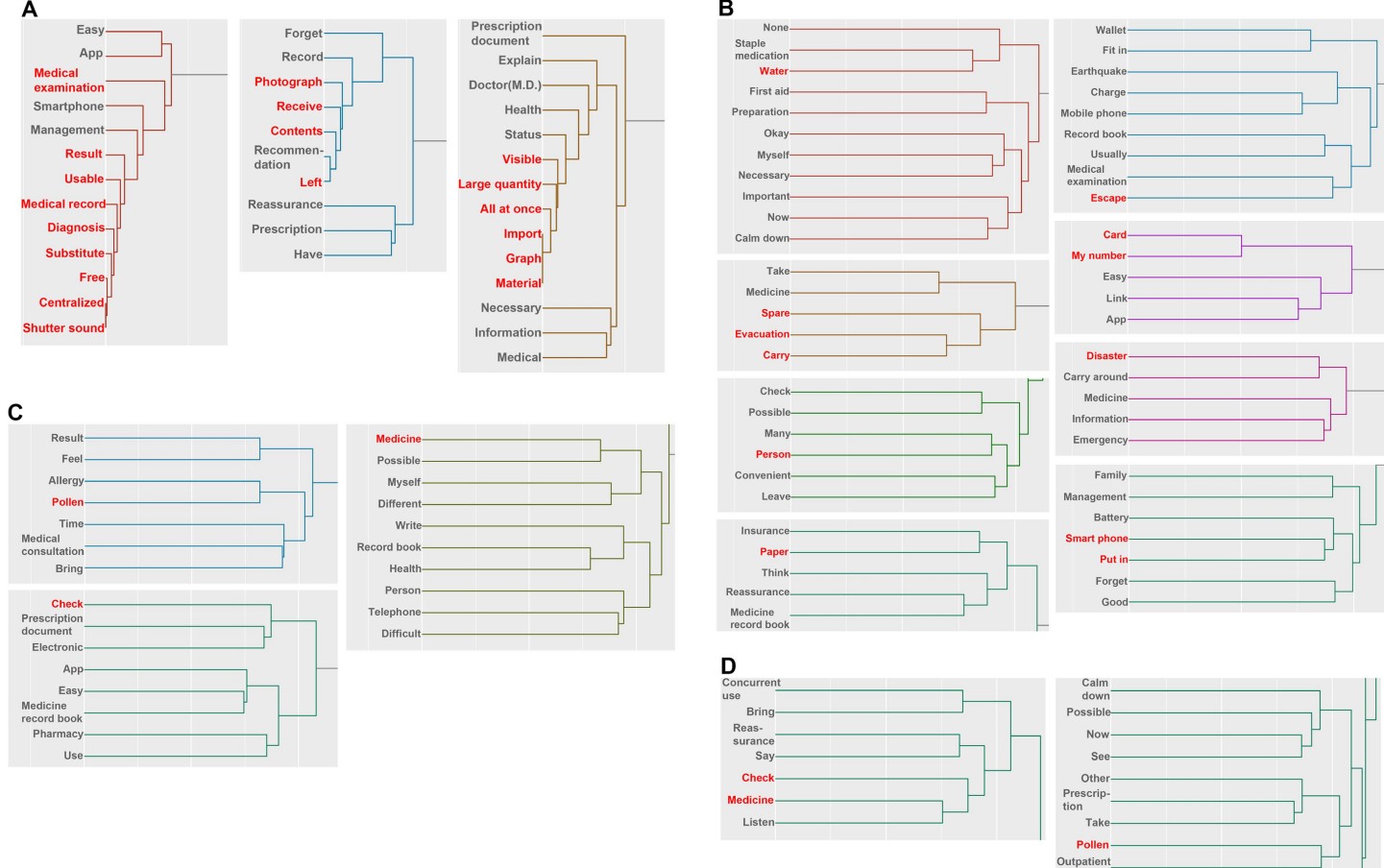

**Fig 4. Hierarchical cluster analysis of posts in which anger during the period from October 2023 to March 2024.** Each clusters shows a strongly related section in the correspondence analysis of posts related to anger. Each figure indicates the results for the following periods: A: October to December, B: January, C: February and D: March. The annotations for conversion from Japanese are shown below. The meaning of Medicine record book is A booklet used to keep a patient's prescription history.

## Conclusion

The results of this study indicate that the psychological impact of the great earthquake on the medical system—specifically the prescription record system—was relatively short-lived. Negative emotions during the disaster were primarily triggered by the unreliability of electronic medical infrastructure, such as the My Number system, following power outages.

However, this study was limited to nationwide trends and did not categorize data by user attributes. Future studies are warranted to examine specific subpopulations, such as medical workers and disaster victims, as this may reveal significant underlying differences not captured in this aggregate analysis. Previous studies have demonstrated that disaster preparedness and healthcare access vary significantly across subpopulations, including socially vulnerable groups [18] and foreign residents [19]. Similarly, social media analysis has revealed distinct behavioral patterns between medical workers and the general public during health crises [20]. Therefore, further research targeting these specific strata is required to develop more inclusive disaster medical strategies.

## Supporting information

**S1 Text. Preparation before analysis by KH coder.** These words were designated for forced extraction to rectify inappropriate segmentation patterns identified during the preliminary analysis using KH Coder.
(PDF)

**S1 Table. Table 1. Results of the preliminary validation of ML-Ask based on a random sample of 400 posts.** The dataset was randomly extracted using a Python script. Two authors independently determined the ground truth labels for the 400 posts. Disagreements were initially addressed through discussion; however, in cases where a consensus could not be reached, both sentiments were adopted as the final ground truth.
(PDF)

**S1 Fig. Hierarchical cluster analysis of posts in which fear during the period from October 2023 to March 2024.** All analysis results are shown. The black boxes highlight areas of strong correlation in the correspondence analysis of posts related to anger. Each figure indicates the results for the following periods: A: October to December, B: January, C: February and D: March.
(TIF)

**S2 Fig. Hierarchical cluster analysis of posts in which dislike during the period from October 2023 to March 2024.** All analysis results are shown. The black boxes highlight areas of strong correlation in the correspondence analysis of posts related to anger. Each figure indicates the results for the following periods: A: October to December, B: January, C: February and D: March.
(TIF)

**S3 Fig. Hierarchical cluster analysis of posts in which anger during the period from October 2023 to March 2024.** All analysis results are shown. The black boxes highlight areas of strong correlation in the correspondence analysis of posts related to anger. Each figure indicates the results for the following periods: A: October to December, B: January, C: February and D: March.
(TIF)

## Author contributions

**Conceptualization:** Tomoya Kitayama.

**Data curation:** Tomoya Kitayama.

**Formal analysis:** Tomoya Kitayama, Kanae Nishimura.

**Investigation:** Tomoya Kitayama, Kanae Nishimura.

**Methodology:** Tomoya Kitayama.

**Project administration:** Tomoya Kitayama.

**Resources:** Tomoya Kitayama.

**Software:** Tomoya Kitayama.

**Supervision:** Tomoya Kitayama.

**Validation:** Tomoya Kitayama, Kanae Nishimura.

**Visualization:** Tomoya Kitayama, Kanae Nishimura.

**Writing – original draft:** Tomoya Kitayama, Kanae Nishimura.

**Writing – review & editing:** Tomoya Kitayama, Kanae Nishimura.

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
