## [Decision Letter · Decision Letter 0]

28 Dec 2025

PONE-D-25-64323The psychological impact of major disasters on Japan's medical system: An SNS text analysisPLOS One

Dear Dr. Kitayama,

Thank you for submitting your manuscript to PLOS ONE. After careful consideration, we feel that it has merit but does not fully meet PLOS ONE’s publication criteria as it currently stands. Therefore, we invite you to submit a revised version of the manuscript that addresses the points raised during the review process.

We look forward to receiving your revised manuscript.

Kind regards,

Liviu-Adrian Cotfas

Academic Editor

PLOS One

Journal Requirements:

2. In your Methods section, please include additional information about your dataset and ensure that you have included a statement specifying whether the collection and analysis method complied with the terms and conditions for the source of the data.

3. Please note that your Data Availability Statement is currently missing the repository name. If your manuscript is accepted for publication, you will be asked to provide these details on a very short timeline. We therefore suggest that you provide this information now, though we will not hold up the peer review process if you are unable.

6.If the reviewer comments include a recommendation to cite specific previously published works, please review and evaluate these publications to determine whether they are relevant and should be cited. There is no requirement to cite these works unless the editor has indicated otherwise.

Additional Editor Comments:

The manuscript must fully comply with PLOS ONE’s publication criteria, particularly with respect to the robustness and support of its conclusions. For instance, the study employs the ML-Ask tool to evaluate emotions; however, the conclusions drawn from this analysis cannot be considered sufficiently supported by the presented results. The manuscript does not assess the performance or validity of ML-Ask on the specific dataset used in this study, leaving uncertainty as to whether the tool performs adequately in this context. Without such an evaluation, the reliability of the emotion analysis—and consequently the conclusions based on it—remains unclear.

Reviewer's Responses to Questions

**Comments to the Author**

1. Is the manuscript technically sound, and do the data support the conclusions?

Reviewer #1: Yes

Reviewer #2: Partly

2. Has the statistical analysis been performed appropriately and rigorously? 

Reviewer #1: Yes

Reviewer #2: Yes

3. Have the authors made all data underlying the findings in their manuscript fully available?

Reviewer #1: Yes

Reviewer #2: Yes

4. Is the manuscript presented in an intelligible fashion and written in standard English?

Reviewer #1: Yes

Reviewer #2: Yes

5. Review Comments to the Author

Reviewer #1: This paper presents an analysis of Japanese tweets mentioning the country's prescription record systems before, right-after and after a major earthquake. Through sentiment and discourse analysis, the authors study the evolution of the topics associated with negative emotions. This works shows that, while the prevalence of emotions didn't change much, the topic associated evolved, with a strong concern about the system's usability right after the earthquake. It also shows that this effect was only short-lived, with pre-earthquake concerns topics returning a month after.

This study is interesting and provides valuable results. Some aspects of the methodology (particularly regarding data collection) should be more detailed, to facilitated reproducibility. To me, the main blind spot of the analysis lies in the tweet's authorship : where these written by users of the prescription records, family of these, health workers, journalists ? A quick qualitative analysis of a subset of the corpus, or a small review of literature about the use of Twitter/X in the Japanese population could lift this ambiguity.

Here follows a list of more precise comments and recommendations :

The introduction clearly describes the topic of prescription records, their implementation in Japan, and the specific challenging that arise during disaster situations. However, the authors do not explain why they choose to analyze it through X/Twitter : is that platform particularly used by health professional ? the population ? what other SNS platform could have been studied ?

The data collection process is presented quickly and some questions remain :

- How is the collection script used to collect tweets older than 7 days ? (I suppose that it was run at regular intervals but this should be explained)

- What API-plan was bought to collect X/Twitter data ?

- What are the limits of the keyword strategy used ? (are they other expressions used to talk about medicine record books ? are some Japanese users tweeting in other languages ?)

- Did the collected dataset require some post-processing or cleaning ?

The sentence "data obtained do not reflect the intent of the authors" is not very-clear and may require explanations (what aspects of the dataset do not meet the authors expectations ?).

The analysis process is described in details. Some general explanations and references on Sentiment analysis could have been provided (here or in the introduction). The section states that occurrences were counted on time periods, but these periods are only described in the result section. Some statistical tests (e.g. Tukey-Kramer) are mentioned but not described ; on the other hand, some procedures (e.g. Cramer's V association coefficient) are described in a lot of details. For a non-specialist like me, this section is therefore difficult to understand. Ethical consideration are clear and useful ; this subsection could be moved just after the one on data collection.

Trends in emotions are presented in detail, the table is very useful. Discussions of the correspondence analysis are clear. The deep dive in the fear, dislike and anger clusters are very interesting. The figures are useful but there is some avoidable redundancy in the paragraphs that follow each figure. It could have been interest, for contrast, to also analyze one or two clusters linked to positive emotions (relief, joy ...).

Conclusions are clear but a bit short. The generalization of this study to other areas and other type of crisis could be discussed. Future research avenues could be proposed.

Reviewer #2: 1. The paper contains conflicting date ranges across sections and figure captions. The Methods section states a collection window from “October 1, 2023 to March 31, 2024,” but multiple figure captions state “October 2024 to March 2025.” This is confusing and difficult to follow. Please make it clear which date range is represented in the data and what date range was used for the analysis.

2. The statistical design is not clearly defined relative to what constitutes an observation unit. The paper reports residual analysis formulas, Cramer’s V, a two-way ANOVA, and a Tukey-Kramer test, but does not define what data points were used for each test. The paper does not specify whether the inputs were daily counts, weekly aggregates, or per-post indicators, or how the denominator for percentages was calculated for each period. The paper reports Cramer’s V as 0.06383 and cites 0.15 as a relevance threshold, but does not justify this threshold choice in the context of this dataset. How exactly were periods and emotions arranged in the contingency table that produced 0.06383, and what were the exact row and column labels?

3. The use of ML Ask as the sentiment and emotion engine is presented as definitive without dataset-specific validation. The paper does not report any manual annotation, interrater agreement, or spot checks on a sampled subset of posts to compare ML Ask labels against human judgment. The paper does not state how negation and sarcasm, common in social media, were handled beyond one illustrative sentence example. The paper reduces downstream analysis to “Fear,” “Dislike,” and “Anger” based on frequency, but does not provide a formal criterion for excluding the remaining 7 emotion axes. The paper also does not report whether multi-label outputs from ML Ask were possible and how those posts were counted in Table 1.

4. A diversity-based analysis is important here because public reactions to medical systems can vary across user groups, and an aggregate trend can hide those differences. The paper does not report any diversity-based patterns, subgroup differences, or user level analysis in the emotion outputs. Several prior works, such as https://doi.org/10.3390/computers12110221 and https://doi.org/10.1016/j.ipm.2021.102541 have highlighted the role of demographic factors when performing similar studies. If performing a diversity-based analysis is not feasible at this point, it is suggested that the authors review a few such works and state this as a future scope of work.

5. The conclusions go beyond what is observed in the analysis. The paper states that negative sentiment toward electronic systems “may lead to a reliance on analog systems” but the paper does not define a measurable indicator of reliance, adoption, or behavioral shift. The paper does not include any triangulation with prescription records usage data, pharmacy-level indicators, or independent survey measures to align the SNS patterns with real system behavior. The paper also does not report whether posts from affected prefectures were distinguished from posts elsewhere in Japan. The paper also does not present a clear mapping from cluster-level terms to specific actionable system failures, such as power supply, authentication, or data access workflows.

6. PLOS authors have the option to publish the peer review history of their article (what does this mean?). If published, this will include your full peer review and any attached files.

Reviewer #1: No

Reviewer #2: No

---

## [Author Response · Author response to Decision Letter 1]

20 Jan 2026

Reply to the reviewers

Thank you very much for your kind letter of December 29, 2025, regarding the constructive comments from the reviewers. We are most grateful for the detailed review. We have gone over the comments carefully and made corrections and additions according to the suggestions. Our point-by-point responses are described below.

Reviewer #1:

This study is interesting and provides valuable results. Some aspects of the methodology (particularly regarding data collection) should be more detailed, to facilitated reproducibility. To me, the main blind spot of the analysis lies in the tweet's authorship : where these written by users of the prescription records, family of these, health workers, journalists ? A quick qualitative analysis of a subset of the corpus, or a small review of literature about the use of Twitter/X in the Japanese population could lift this ambiguity.

The primary purpose of this study was to examine nationwide social trends at the time of the earthquake, and unfortunately, a detailed analysis of subpopulations was not feasible with the current dataset. Therefore, we have revised the Conclusion section to explicitly state this limitation and the need for future research. The updated text is provided below.

→P2 Line 29

These results suggest that during the earthquake in Japan, a significant number of people harbored negative sentiments toward electronic medical systems and questioned their reliability in disaster situations.

→P28 Line 414

The results of this study indicate that the psychological impact of the great earthquake on the medical system—specifically the prescription record system—was relatively short-lived. Negative emotions during the disaster were primarily triggered by the unreliability of electronic medical infrastructure, such as the My Number system, following power outages.

However, this study was limited to nationwide trends and did not categorize data by user attributes. Future studies are warranted to examine specific subpopulations, such as medical workers and disaster victims, as this may reveal significant underlying differences not captured in this aggregate analysis. Previous studies have demonstrated that disaster preparedness and healthcare access vary significantly across subpopulations, including socially vulnerable groups [18] and foreign residents [19]. Similarly, social media analysis has revealed distinct behavioral patterns between medical workers and the general public during health crises [20]. Therefore, further research targeting these specific strata is required to develop more inclusive disaster medical strategies.

The introduction clearly describes the topic of prescription records, their implementation in Japan, and the specific challenging that arise during disaster situations. However, the authors do not explain why they choose to analyze it through X/Twitter : is that platform particularly used by health professional ? the population ? what other SNS platform could have been studied ?

→P7 Line 86, I added the following sentences:

X was selected as the subject of analysis due to its widespread usage across diverse generations and its high penetration rate in Japan [6]. While other platforms such as LINE, YouTube, Instagram, and TikTok also maintain large user bases in Japan, they were excluded from this study. LINE was omitted due to its nature as a private communication tool, which presents privacy constraints. The other platforms are primarily video- or image-oriented, rendering them unsuitable for the text mining methods employed in this research.

The data collection process is presented quickly and some questions remain :

- How is the collection script used to collect tweets older than 7 days ? (I suppose that it was run at regular intervals but this should be explained)

- What API-plan was bought to collect X/Twitter data ?

→P7 Line 97, I added the following sentences:

The program was implemented by the authors, who rotated roles every one weeks. It should be noted that due to limitations of the Twitter API (X API Basic) and Internet environment, not all relevant posts were collected.

- What are the limits of the keyword strategy used ? (are they other expressions used to talk about medicine record books ? are some Japanese users tweeting in other languages ?)

Despite the lack of a legal definition, the term " medicine record book (Okusuri Techo)" is promoted by the MHLW and universally recognized across Japan. Consequently, the term is regarded as stable, with little risk of orthographical fluctuation affecting the analysis. https://www.mhlw.go.jp/content/001332913.pdf

- Did the collected dataset require some post-processing or cleaning ?

Regarding the pre-processing of the data, we strictly limited the cleaning to the removal of URLs and tags. We intentionally retained punctuation marks and symbols (e.g., exclamation marks) because they often convey strong sentiment and are essential for the accuracy of the emotion recognition engine used in this study. The updated text is provided below.

→P10 Line 144

The resulting data were generated as .csv files using Python. Some of the posts contained URL and tag (ex; @aaaaa) information that had no meaning in Japanese, and preprocessing was performed to remove them, which prevented analytical errors [7]. Since the trial analysis by KH Coder detected words separated by inappropriate delimiters, forced extraction words were set (S1 Text).

The sentence "data obtained do not reflect the intent of the authors" is not very-clear and may require explanations (what aspects of the dataset do not meet the authors expectations ?).

→P7 Line 99, I edited the following sentences:

Data collection was automated using a Python program, ensuring that the process was mechanical and independent of the authors' subjective intent.

The analysis process is described in details. Some general explanations and references on Sentiment analysis could have been provided (here or in the introduction). The section states that occurrences were counted on time periods, but these periods are only described in the result section. Some statistical tests (e.g. Tukey-Kramer) are mentioned but not described ; on the other hand, some procedures (e.g. Cramer's V association coefficient) are described in a lot of details. For a non-specialist like me, this section is therefore difficult to understand. Ethical consideration are clear and useful ; this subsection could be moved just after the one on data collection.

→P13 Line 193, I edited the following sentences:

Multiple comparison tests for the frequency of posts containing each emotion were performed using GraphPad Prism 6.04 (GraphPad Software, San Diego, CA, USA). Data are presented as the mean ± standard deviation (SD). The frequency of posts was analyzed using a two-way analysis of variance (ANOVA) with Emotion and Period as independent factors. Following the ANOVA, a Tukey-Kramer post-hoc test was conducted to identify specific differences between groups. A p-value of < 0.05 was considered statistically significant.

→Moved the Ethics subsection. P8 Line 106

Trends in emotions are presented in detail, the table is very useful. Discussions of the correspondence analysis are clear. The deep dive in the fear, dislike and anger clusters are very interesting. The figures are useful but there is some avoidable redundancy in the paragraphs that follow each figure. It could have been interest, for contrast, to also analyze one or two clusters linked to positive emotions (relief, joy ...).

We appreciate your suggestion. In fact, we had assessed the validity of the emotion recognition engine prior to our initial submission, although these results were not included in the original manuscript. In this revision, we have added these assessment results in response to a comment from another reviewer. As these results indicate that positive emotions (such as relief and joy) have significantly lower reliability and frequency compared to negative ones, we believe a direct comparison is difficult to justify.

Conclusions are clear but a bit short. The generalization of this study to other areas and other type of crisis could be discussed. Future research avenues could be proposed.

I edited the following sentences:

→P2 Line 29

These results suggest that during the earthquake in Japan, a significant number of people harbored negative sentiments toward electronic medical systems and questioned their reliability in disaster situations.

→P28 Line 414

The results of this study indicate that the psychological impact of the great earthquake on the medical system—specifically the prescription record system—was relatively short-lived. Negative emotions during the disaster were primarily triggered by the unreliability of electronic medical infrastructure, such as the My Number system, following power outages.

However, this study was limited to nationwide trends and did not categorize data by user attributes. Future studies are warranted to examine specific subpopulations, such as medical workers and disaster victims, as this may reveal significant underlying differences not captured in this aggregate analysis. Previous studies have demonstrated that disaster preparedness and healthcare access vary significantly across subpopulations, including socially vulnerable groups [18] and foreign residents [19]. Similarly, social media analysis has revealed distinct behavioral patterns between medical workers and the general public during health crises [20]. Therefore, further research targeting these specific strata is required to develop more inclusive disaster medical strategies.

Reviewer #2

1. The paper contains conflicting date ranges across sections and figure captions. The Methods section states a collection window from “October 1, 2023 to March 31, 2024,” but multiple figure captions state “October 2024 to March 2025.” This is confusing and difficult to follow. Please make it clear which date range is represented in the data and what date range was used for the analysis.

→Thank you for your advice. I checked the content and corrected it.

2. The statistical design is not clearly defined relative to what constitutes an observation unit. The paper reports residual analysis formulas, Cramer’s V, a two-way ANOVA, and a Tukey-Kramer test, but does not define what data points were used for each test. The paper does not specify whether the inputs were daily counts, weekly aggregates, or per-post indicators, or how the denominator for percentages was calculated for each period. The paper reports Cramer’s V as 0.06383 and cites 0.15 as a relevance threshold, but does not justify this threshold choice in the context of this dataset. How exactly were periods and emotions arranged in the contingency table that produced 0.06383, and what were the exact row and column labels?

→P12 Line 172 and P13 Line 190, I added the following sentences:

→P12 Line 172

To account for variations in the total number of posts across different periods, we calculated the percentage of posts containing each specific emotion. The percentage was defined as: (number of posts with a specific emotion / total number of posts in the period) × 100. The study period was divided into four intervals: October to December, January, February, and March.

→P13 Line 190

The number of categories was determined based on the smaller dimension. Therefore, a value of 4, corresponding to the number of periods (columns) in Table 1, was selected rather than the number of emotions (rows).

3. The use of ML Ask as the sentiment and emotion engine is presented as definitive without dataset-specific validation. The paper does not report any manual annotation, interrater agreement, or spot checks on a sampled subset of posts to compare ML Ask labels against human judgment. The paper does not state how negation and sarcasm, common in social media, were handled beyond one illustrative sentence example. The paper reduces downstream analysis to “Fear,” “Dislike,” and “Anger” based on frequency, but does not provide a formal criterion for excluding the remaining 7 emotion axes. The paper also does not report whether multi-label outputs from ML Ask were possible and how those posts were counted in Table 1.

We appreciate your suggestion. We had actually assessed the validity of the emotion recognition engine prior to our initial submission, although these results were not included in the original manuscript. In this revision, we have added the details of this assessment.

→P10 Line 149

Dataset-specific validation of ML-Ask was performed using random sampling. We determined the minimum sample size using the Raosoft sample size online calculator (http://www.raosoft.com/samplesize.html; Raosoft Inc., Seattle, WA, USA) based on a total population of 38,567 posts, assuming a 5% margin of error and a 95% confidence level. Although the required sample size was 381, we randomly selected 400 samples using a Python script (https://github.com/tomokitamukogawa/Psychological-impact-of-disasters-on-system) to ensure robustness.

Each of the 400 extracted sentences was evaluated independently by the authors. Disagreements were initially addressed through discussion to determine the final ground truth label. However, in cases where a consensus could not be reached, both sentiments identified by the authors were adopted as valid ground truth labels. Since ML-Ask supports multi-label classification, we constructed a 2×2 contingency table for each emotion and calculated the Cohen’s kappa coefficient for each, as well as the macro-average (S1 Table). All calculations were performed using Python (https://github.com/tomokitamukogawa/Psychological-impact-of-disasters-on-system).

The strength of agreement was interpreted according to the criteria of Landis and Koch (1977): Slight (0.00–0.20), Fair (0.21–0.40), Moderate (0.41–0.60), Substantial (0.61–0.80), and Almost Perfect (0.81–1.00)[15]. The macro-average kappa coefficient across all emotions was 0.449, indicating "moderate" agreement [16]. Notably, Anger, Dislike, Fear, and Shame showed particularly high agreement, with kappa coefficients exceeding 0.6 (S1 Table). It should be noted that emotions with lower prevalence tended to yield lower kappa coefficients.

4. A diversity-based analysis is important here because public reactions to medical systems can vary across user groups, and an aggregate trend can hide those differences. The paper does not report any diversity-based patterns, subgroup differences, or user level analysis in the emotion outputs. Several prior works, such as https://doi.org/10.3390/computers12110221 and https://doi.org/10.1016/j.ipm.2021.102541 have highlighted the role of demographic factors when performing similar studies. If performing a diversity-based analysis is not feasible at this point, it is suggested that the authors review a few such works and state this as a future scope of work.

5. The conclusions go beyond what is observed in the analysis. The paper states that negative sentiment toward electronic systems “may lead to a reliance on analog systems” but the paper does not define a measurable indicator of reliance, adoption, or behavioral shift. The paper does not include any triangulation with prescription records usage data, pharmacy-level indicators, or independent survey measures to align the SNS patterns with real system behavior. The paper also does not report whether posts from affected prefectures were distinguished from posts elsewhere in Japan. The paper also does not present a clear mapping from cluster-level terms to specific actionable system failures, such as power supply, authentication, or data access workflows.

The response to comments 4 and 5 is as follows.

The primary purpose of this study was to examine nationwide social trends at the time of the earthquake, and unfortunately, a detailed analysis of subpopulations was not feasible with the current dataset. Therefore, we have revised the Conclusion section to explicitly state this limitation and the need for future research. The updated text is provided below.

→P2 Line 29

These results suggest that during the earthquake in Japan, a significant number of people harbored negative sentiments toward electronic medical syst

---

## [Decision Letter · Decision Letter 1]

1 Feb 2026

The psychological impact of major disasters on Japan's medical system: An SNS text analysis

PONE-D-25-64323R1

Dear Dr. Kitayama,

We’re pleased to inform you that your manuscript has been judged scientifically suitable for publication and will be formally accepted for publication once it meets all outstanding technical requirements.

Kind regards,

Liviu-Adrian Cotfas

Academic Editor

PLOS One

Additional Editor Comments (optional):

Reviewers' comments:

Reviewer's Responses to Questions

**Comments to the Author**

1. If the authors have adequately addressed your comments raised in a previous round of review and you feel that this manuscript is now acceptable for publication, you may indicate that here to bypass the “Comments to the Author” section, enter your conflict of interest statement in the “Confidential to Editor” section, and submit your "Accept" recommendation.

Reviewer #1: All comments have been addressed

Reviewer #2: All comments have been addressed

2. Is the manuscript technically sound, and do the data support the conclusions?

Reviewer #1: Yes

Reviewer #2: (No Response)

3. Has the statistical analysis been performed appropriately and rigorously? 

Reviewer #1: Yes

Reviewer #2: (No Response)

4. Have the authors made all data underlying the findings in their manuscript fully available?

Reviewer #1: Yes

Reviewer #2: (No Response)

5. Is the manuscript presented in an intelligible fashion and written in standard English?

Reviewer #1: Yes

Reviewer #2: (No Response)

6. Review Comments to the Author

Reviewer #1: (No Response)

Reviewer #2: (No Response)

7. PLOS authors have the option to publish the peer review history of their article (what does this mean?). If published, this will include your full peer review and any attached files.

Reviewer #1: No

Reviewer #2: No

---

## [Editor Report · Acceptance letter]

PONE-D-25-64323R1

PLOS One

Dear Dr. Kitayama,

I'm pleased to inform you that your manuscript has been deemed suitable for publication in PLOS One. Congratulations! Your manuscript is now being handed over to our production team.

Kind regards,

on behalf of

Dr. Liviu-Adrian Cotfas

Academic Editor

PLOS One